# Face selective patches in marmoset frontal cortex

David J. Schaeffer [1✉], Janahan Selvanayagam [2], Kevin D. Johnston[2], Ravi S. Menon[1], Winrich A. Freiwald[3,4] & Stefan Everling[1,2]

In humans and macaque monkeys, socially relevant face processing is accomplished via a distributed functional network that includes specialized patches in frontal cortex. It is unclear whether a similar network exists in New World primates, who diverged ~35 million years from Old World primates. The common marmoset is a New World primate species ideally placed to address this question given their complex social repertoire. Here, we demonstrate the existence of a putative high-level face processing network in marmosets. Like Old World primates, marmosets show differential activation in anterior cingulate and lateral prefrontal cortices while they view socially relevant videos of marmoset faces. We corroborate the locations of these frontal regions by demonstrating functional and structural connectivity between these regions and temporal lobe face patches. Given the evolutionary separation between macaques and marmosets, our results suggest this frontal network specialized for social face processing predates the separation between Platyrrhini and Catarrhini.

[1] Centre for Functional and Metabolic Mapping, Robarts Research Institute, University of Western Ontario, London, ON N6A 5K8, Canada. [2] Department of Physiology and Pharmacology, University of Western Ontario, London, ON N6A 5K8, Canada. [3] Laboratory of Neural Systems, The Rockefeller University, New York, NY 10065, USA. [4] The Center for Brains, Minds and Machines, MIT, Cambridge, MA 02139, USA. ✉email: dschaeff@uwo.ca

The circuitry responsible for face processing has been well documented in humans and macaques[1–5]. Old World primate species seem to share a common architecture of this circuitry, with multiple face-selective patches along the occipitotemporal axis that are functionally connected with a larger face processing network that includes several subcortical areas (e.g., hippocampus and amygdala[6]) and face-selective patches in frontal cortex[1,7]. The face-selective patches in frontal cortex have been implicated in processing social context and orofacial movements in macaques—anterior cingulate cortex and lateral prefrontal cortex are differentially activated when in direct visual contact with the face of a conspecific[8,9]. It is unclear whether a similar network exists in New World primates, who separated ~35 million years ago from Old World primates[10]. The common marmoset (*Callithrix jacchus*) is a small New World primate that is ideally placed to address this question given the rich social repertoire inherent to this species (e.g., observational social learning; imitation; cooperative antiphonal calling[11]). Marmosets, however, have a less elaborated frontal cortex when compared to Old World primate species including macaques[12]. Here, we used ultrahigh field (9.4 Tesla) task-based functional magnetic resonance imaging (fMRI) in marmosets to investigate whole brain face processing in marmosets.

Recent fMRI studies have demonstrated that marmosets do indeed possess face patches in a ventral pathway along the temporal lobes that have a similar organization to Old World primates[13,14]. Given that marmosets use eye contact and facial expression as a means of social communication[15–17], we posited that marmosets could also possess a face processing network that extends into frontal cortex. The frontal constituents of face-to-face interaction in macaques have been demonstrated by showing conspecific videos during fMRI acquisition[8]—when viewing videos of other macaques in a "direct-gaze" context (i.e., simulated eye contact) a patch of anterior cingulate cortex is differentially activated. Interestingly, this patch is less active when viewing videos of other macaques in an "averted-gaze" context (i.e., while the monkey in the video is looking away). Here, our goal was to employ a marmoset conspecific version of this task during whole brain fMRI to test for the existence of functional face patches in frontal cortex.

By leveraging our recent hardware advances in ultrastable awake marmoset imaging[18], we acquired whole brain fMRI in four marmosets while they viewed videos of marmoset faces in social (with directed or averted gaze) or nonsocial (scrambled videos) conditions. To quantify gaze differences between the stimuli, we also performed eye tracking in five marmosets while they performed the same task outside of the MRI environment. To corroborate the connectivity of the task-based circuitry, we utilized our extensive fully awake resting-state fMRI (RS-fMRI) dataset to index functional connectivity. To index structural connectivity, we overlaid the results of tracer-based cellular connectivity data of multiple anterior cingulate cortex injection sites in marmosets[19]. Based on these data we demonstrate the existence of a putative high-level face processing network in marmosets. Like Old World primates, marmosets show differential activation in anterior cingulate and lateral prefrontal cortices while they view socially relevant videos of marmoset faces. Given the evolutionary separation between macaques and marmosets, our results suggest this frontal network specialized for social face processing predates the separation between Platyrrhini and Catarrhini.

## Results
### Task-based fMRI comparisons.
Figure 1 shows group maps comparing the social video conditions (including both directed and averted gaze) and nonsocial scrambled versions of those videos. As shown in Fig. 1, videos containing conspecific faces elicited a broad network that was similar, but stronger than the topology that was elicited using the scrambled versions of the videos. The social conditions showed stronger activation along the occipitotemporal axis with peaks in V4/TEO and TE3. In frontal cortex, the social videos showed peaks laterally in 45/47 L and orbitofrontally in 13 L (albeit orbitofrontal cortex suffered from relatively low signal-to-noise ratio, see Schaeffer et al., 2019a). Along the medial cortical surface, the largest differences were present in visual cortex (V1 and V2). We also found subcortical differences between the social and nonsocial scrambled conditions, including in the superior colliculus (SC), hippocampus (Hipp), Pulvinar (Pul), medial-dorsal nucleus of thalamus (MD), and in the amygdala (Amy).

Figure 2 shows the comparison between the two different social conditions—directed gaze and averted gaze. Overall, the broad topologies of these circuitries were similar, but Fig. 2c shows several critical differences when contrasting the two conditions. The activation was greater for the directed gaze videos along the occipitotemporal axis—these regions are remarkably similar to the face patches identified in marmosets by Hung et al. (2015). As such, we have adopted the terminology used in their manuscript: occipital (O; V2/V3), posterior ventral (PV; V4/TEO), posterior dorsal (PD; FST), middle dorsal (MD; caudal TE), and anterior dorsal (AD; rostral TE). In addition to these face-selective patches in temporal lobe, we also found clear peaks related to the directed gaze videos in anterior cingulate (at the confluence of 8b, 32, and 24) and lateral frontal cortex (45/47 L). Subcortically, SC, Pul, Hipp, and MD also showed face selectivity when contrasting the directed and averted-gaze conditions.

### Resting-state seed analysis.
Considering the findings regarding the frontal face patches in the task-based experiment described above (i.e., anterior cingulate and lateral prefrontal cortex differentially activating for directed gaze faces), we sought to corroborate the functional connectivity between the temporal face patches and those found in frontal cortex. As such, we calculated functional connectivity between the face patches (AD, MD, PD, and PV) and the rest of the brain as shown in Fig. 3a. Indeed, AD, MD, and PD all connected (with varying extents) to the 8b/24 anterior cingulate cluster and the lateral frontal cortex cluster (45/47). Generally, the anterior face patches (MD and AD) connected most strongly with anterior cingulate cortex, whereas as the more posterior face patches (PV and PD) strongly connected to lateral frontal cortex. PV connected strongly with lateral frontal cortex, but not anterior cingulate cortex. When the frontal patches were seeded, 8b/24 showed peaks of connectivity near the anterior faces patches (MD and AD), whereas 45/47 L showed strong connectivity across both the anterior (MD and AD) and the posterior (PD and PV) face patches.

### Comparison with tracer-based cellular connectivity.
With cortical tracer injections publicly available[19] we were also able to compare our findings with structural connectivity in marmosets. As shown in Fig. 3b, injections proximal to the face-selective anterior cingulate cortex cluster (8b/24) show clear connectivity with lateral frontal cortex (45/47). Further, consonant with our functional connectivity analysis, anterior cingulate cortex injections show strong connectivity with the anterior face patches (AD and MD), but weaker connectivity with the posterior face patches (PD and PV). As shown in Fig. 3c, injections into area 45/47 (particularly CJ800-CTbgr) show very strong connectivity with both anterior cingulate cortex and also along the occipitotemporal pathway harboring the face patches. Accordingly, as

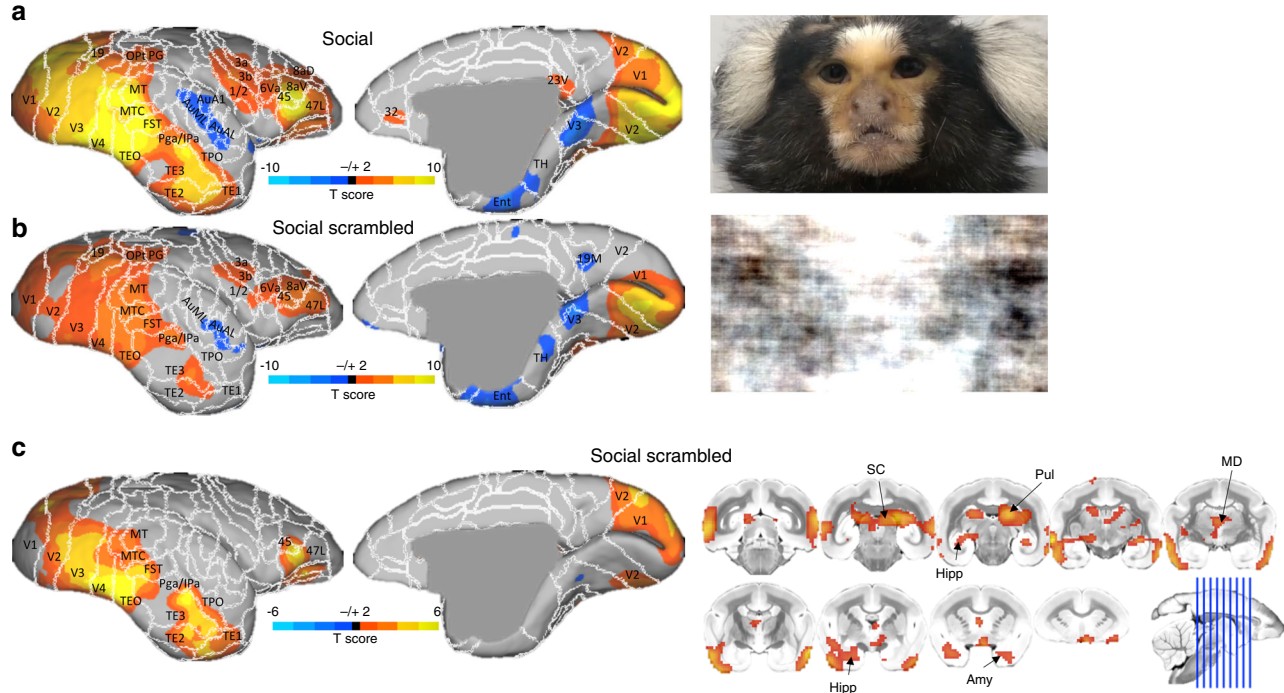

**Fig. 1 Face processing topologies.** Group functional topology comparisons between social (i.e., directed and averted gaze) and nonsocial scrambled video conditions displayed on a marmoset cortical surface. **a** shows group topologies for social videos and **b** shows topologies for social scrambled videos, with stills of representative stimuli to the right of each surface. **c** shows the contrast between the social and scrambled conditions, with volumetric display (to show subcortical activation) to the right of the medial and lateral surfaces.

also suggested by the resting-state seed analysis above, the 8b/24 patch is strongly connected to the anterior faces patches (AD and MD) whereas the lateral patch (45/47 L) is more broadly connected across the face patches.

**Eye tracking.** Distributions representing proportion of saccades by saccade amplitude and proportion of fixations by duration are shown in Fig. 4. Separate repeated measure ANOVAs were performed for saccade amplitudes and fixation durations with the factor of conditions (four levels: fixation, scrambled face, directed gaze, and averted gaze). For saccade amplitudes, a significant effect of condition was observed, $F(3,12) = 8.02$, $p = 0.00$, $MSE = 0.23$, $\eta_p^2 = 0.67$, where saccade amplitudes were significantly longer in the averted-gaze condition than in the fixation or scrambled face conditions. However, no significant differences between conditions were observed after correcting for multiple comparisons. Similarly for fixation durations, a significant effect of condition was observed, $F(3,12) = 4.98$, $p = 0.02$, $MSE = 86.28$, $\eta_p^2 = 0.55$, where fixation duration was longer for the fixation condition than for the averted-gaze condition but pairwise comparisons revealed no significant differences between conditions after corrections. Overall, these results suggest that the cortical topologies were likely not directly driven by differences in saccade number or amplitudes.

**Discussion**
In this study, we were interested in determining whether New World marmosets show face-selective patches in frontal cortex, as has been demonstrated in Old World macaques[1,7,8]. To do so, we presented videos of marmoset faces during fMRI acquisition at ultrahigh field. Similar to macaques, marmosets showed differential activation in anterior cingulate cortex and lateral frontal cortex when viewing videos of conspecific faces with directed gaze (i.e., direct eye contact) when compared to averted-gaze videos[8].

To corroborate the connectivity of these frontal face patches, we also compared our task-based fMRI results with RS-fMRI based functional connectivity and tracer-based structural connectivity. Both analyses demonstrated strong connectivity between the temporal face patches and the frontal face patches, with the anterior and posterior patches differentially connected to medial (8b/24) and lateral frontal cortex (45/47), respectively. Overall, these findings suggest that marmosets do indeed possess a face processing circuitry that extends into frontal cortex and likely supports socially relevant processing of faces.

When comparing patterns of activation elicited from videos of marmoset faces, our results are remarkably similar to those shown in a previous marmoset fMRI study[13], which presented photos of marmoset faces, bodies, objects, or scrambled versions of these photos and found face selectivity in six occipitotemporal patches (O, PD, PV, MD, AD, and MV). We corroborate their results by eliciting all of these patches with marmoset face videos, with the exception of MV—in fact, this group also did not see this patch with fMRI acquisition, but rather used electrocorticography arrays to index activity in this ventrolateral region. Here, we likely also did not see MV in our topologies because of the low signal-to-noise ratio in this area (see ref. [18] for receive coil design and signal-to-noise topologies).

The temporal face-selective patches in marmosets shown here are comparable to those found in humans and macaques (as reviewed in ref. [3]). In humans, these regions exist along the ventral temporal surface, which is a slightly different organization than in macaques, which show patches that are more dorsally located, around the superior temporal sulcus[1]. These Old World primate species, however, also show face-selective patches in frontal cortex—to date, this had yet to be demonstrated in New World marmosets. Here, as shown in Fig. 2, we demonstrate that marmosets do indeed show face selectivity in anterior cingulate and lateral frontal cortex. These patterns were obtained by contrasting patterns of activation elicited from videos of marmosets

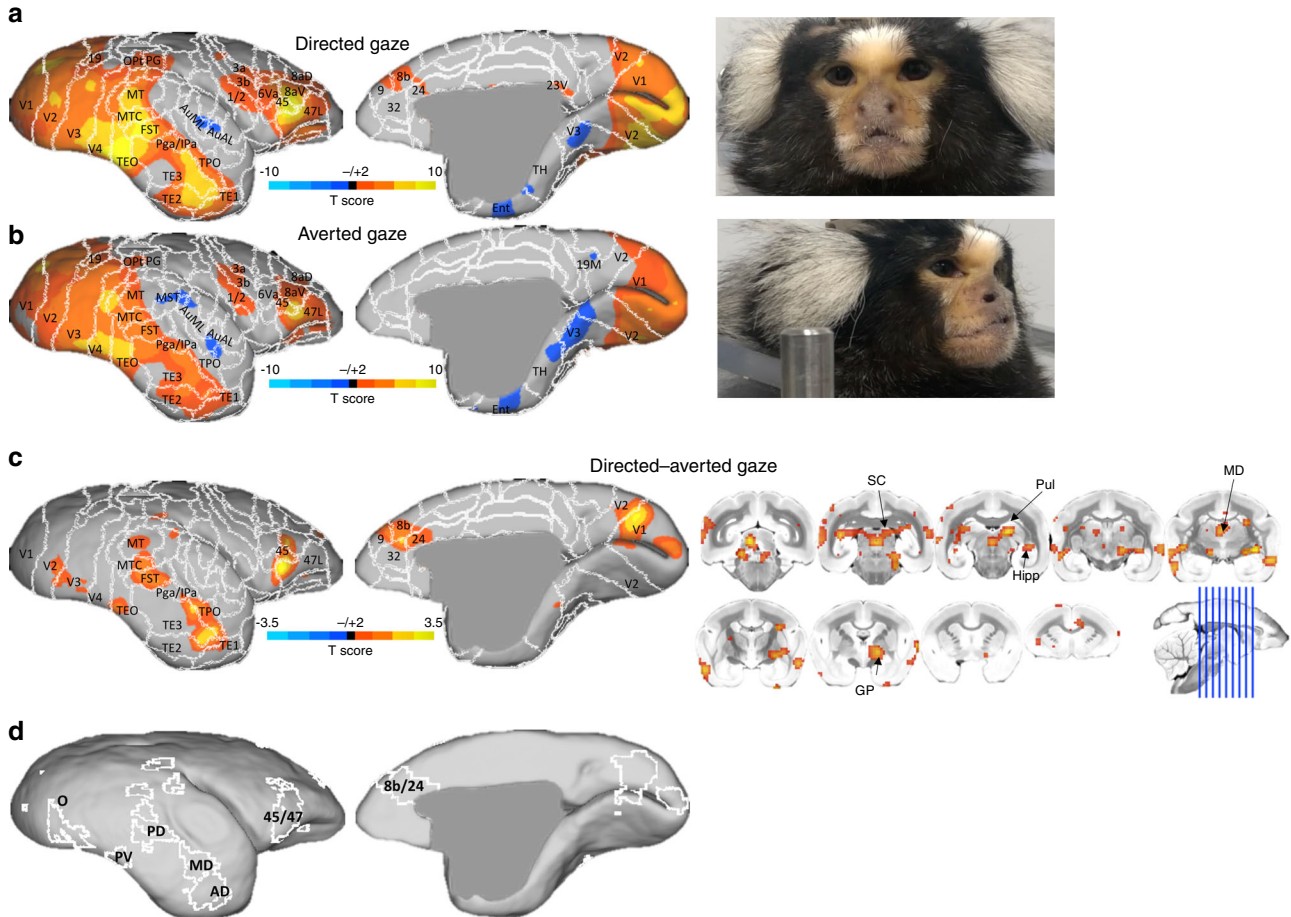

**Fig. 2 Social face processing topologies.** Group functional topology comparisons between directed and averted-gaze video conditions displayed on a marmoset cortical surface. **a** shows group topologies for directed gaze videos and **b** shows topologies for averted-gaze videos, with stills of representative stimuli to the right of each surface. **c** shows the contrast between the directed and averted-gaze conditions, with volumetric display (to show subcortical activation) to the right of the medial and lateral surfaces. **d** shows the outline of the pattern from **c** with the addition of labels for the face-selective patches.

with directed gaze (Fig. 2a) versus marmosets looking toward the left or right (i.e., averted gaze; Fig. 2b). As previously demonstrated in macaques (using a similar conspecific video set), the patch in anterior cingulate cortex (8b/24) was specific to the directed gaze condition, suggesting that this patch is involved in socially relevant face processing[8]. The location of this patch is also in-line with what is found in macaques, residing just dorsal to area 32, extending into areas 24, 8, and 9[6,8].

The lateral prefrontal patch (peak between 45 and 47) shown here, however, was elicited in both directed and averted conditions, albeit to a stronger degree in the directed gaze condition. We hypothesize that the dorsal part of this functional cluster, especially that reaching into area 8aV, is related to small eye movements (i.e., making saccades to salient features of the videos), rather than being face-specific per se. The lateral portion of this cluster, however, is likely face specific—we have recently demonstrated topologies related to saccadic eye movements in marmosets using both microsimulation[20] and task-based fMRI[21] —when directly overlaying these patterns, it seems that the peak in area 45/47 shown in Fig. 2c is likely too far ventrolateral to be related to eye movements. Further, neither the saccade amplitude nor fixation duration differed between these conditions (Fig. 4). Therefore, this patch in lateral frontal cortex is likely face selective. Given that this patch is present in both macaques and humans, we hypothesize that this patch could be the marmoset homolog to these regions[1,6–8].

Evidence for face selectivity in anterior cingulate cortex and lateral frontal cortex patches was further substantiated by both functional and structural connectivity between these regions and the temporal face patches. To index functional connectivity, we utilized our ultrastable (~150 µm maximum head motion), fully awake RS-fMRI dataset acquired at 9.4 Tesla. We seeded four of the face patches (AD, MD, PD, and PV, but did not seed O due to concerns with low signal, see ref. [18] on this issue) and found strong connectivity between AD, MD, and PD with the anterior cingulate cortex face patch (Fig. 3a). Interestingly, the anterior face patches (MD and AD) connected most strongly with anterior cingulate cortex, whereas the more posterior face patches (PV and PD) strongly connected to lateral frontal cortex. This pattern was also present when we examined tracer-based structural connectivity (Fig. 3b), with two separate retrograde tracer injections proximal to the anterior cingulate face patch showing clear connectivity with the AD and MD patches. Further, these anterior cingulate cortex injections also showed strong connectivity with the lateral frontal face-selective patch elicited via the directed gaze videos. Taken together, these results suggest that the anterior cingulate cortex and lateral frontal cortex face patches are part of a cortical face processing network in the marmoset brain.

Our results also provide evidence for the subcortical constituents of a face processing network in marmosets. With our custom cortex-oriented receive coil[18] we were not equipped to be maximally sensitive to subcortical regions, but the general pattern

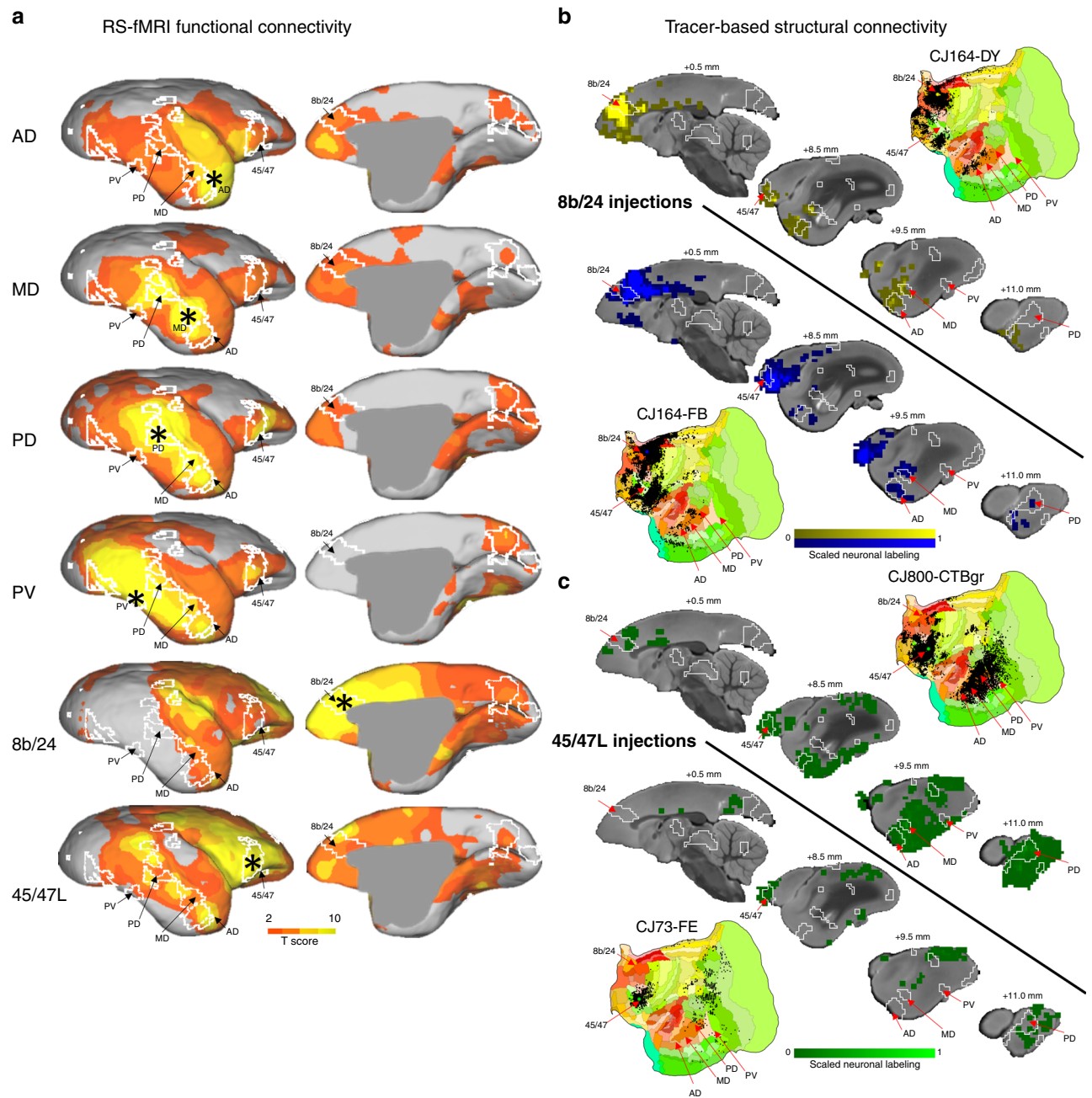

**Fig. 3 Functional and structural connectivity of marmoset face patches. a** shows RS-fMRI based functional connectivity of four temporal and two frontal face patches with the rest of the brain. **b** shows the results of retrograde tracer injections, which were proximal to our anterior cingulate cortex face patch. **c** shows the results of retrograde tracer injections, which were proximal to our 45/47 L face patch. Surface flat map representations of these tracer injections (downloaded from marmosetbrain.org) are also shown. In all panels **a–c** white lines show the functional clusters found by comparing the directed and averted-gaze conditions for reference (i.e., the topology shown in Fig. 2d).

found here is quite similar to that shown in Old World primates[6,22–25]. When contrasting the social and nonsocial scrambled conditions, we found differential activation in SC, Hipp, Pul, MD, and Amy. Face selectivity in regions such as SC, Pul, Hipp and Amy is intriguing and our findings certainly warrant further investigation on this subject, perhaps with more sensitive electrophysiological techniques that have been used to show subcortical face selectivity in macaques[24,26]. The fMRI results here may serve as a starting block for these more invasive explorations of face-selective patches across the marmoset brain.

In summary, we report evidence from task-based fMRI, RS-fMRI, and tracer-based structural connectivity for a face

processing network in New World marmosets, which includes at least two face-selective patches in frontal cortex. We demonstrated that, as in Old World primates, these frontal face patches are differentially sensitive to social interaction (i.e., direct eye contact), suggesting a high-level, top-down control social processing network in the marmoset brain. Therefore, our results suggest that the origin of this frontal network specialized for social face processing predates the separation between Old and New World primates around 35 million years ago[10]. These results give further credence to the marmoset as a viable preclinical modelling species for studying human social disorders.

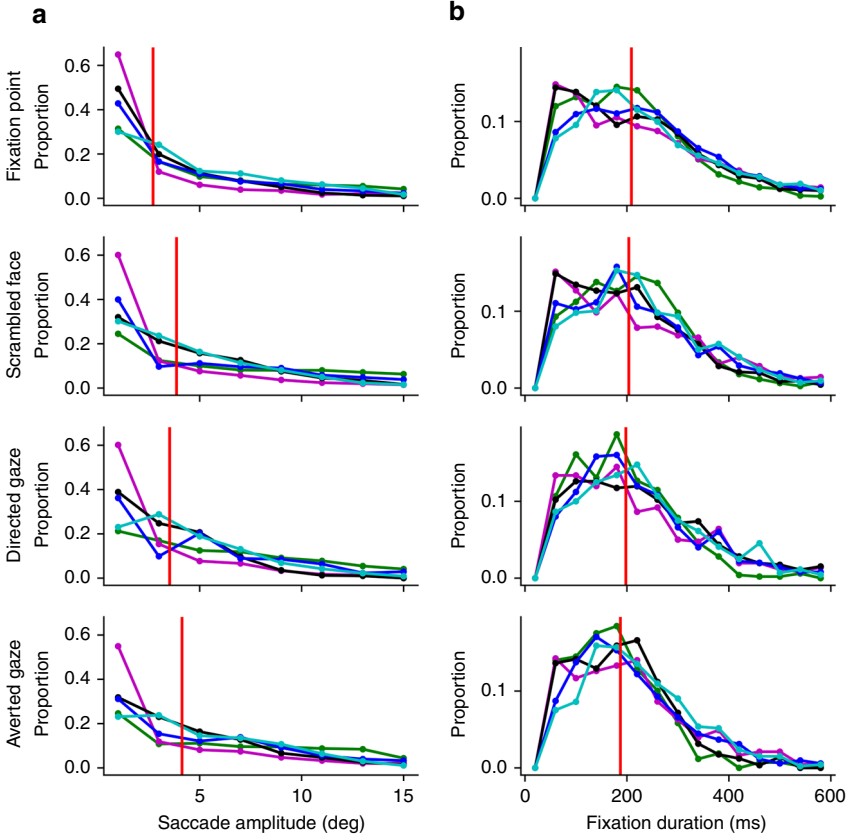

**Fig. 4 Saccadic eye movements.** Distributions of saccades by saccade amplitude in visual degrees **a** and fixations by fixation duration in ms **b** for each condition separately for five marmoset subjects. Different line colors denote individual marmoset subjects. Vertical red lines represent the group median value. Source data are provided as a Source Data file.

## Methods

**Subjects.** Data were collected from 10 adult marmosets (*Callithrix jacchus*; three female; weight 245–380 g; age 29–74 months), with $n = 4$ for task-based fMRI, $n = 5$ for RS-fMRI, and $n = 5$ for the eye tracking outside of the MRI (all but 1 exclusive to the monkeys used in fMRI experiments). Experimental procedures were in accordance with the Canadian Council of Animal Care policy and a protocol approved by the Animal Care Committee of the University of Western Ontario Council on Animal Care. All animal experiments complied with the ARRIVE guidelines.

**Marmoset surgical implantation and head-fixation training.** All 10 marmosets underwent an aseptic surgical procedure to implant a head chamber, five of which were MRI-compatible (i.e., implanted using non-radio opaque dental cement). The purpose of the chamber was to fix the head and thereby prevent animal motion during MRI acquisition. For the chamber implantation procedure[18,27], several coats of adhesive resin (All-bond Universal, Bisco, Schaumburg, Illinois, USA) were applied using a microbrush, air dried, and cured with an ultraviolet dental curing light (King Dental). Then, a two-component dental cement (C & B Cement, Bisco, Schaumburg, Illinois, USA) was applied to the skull and to the bottom of the chamber, which was then lowered onto the skull via a stereotactic manipulator to ensure correct location and orientation. The chamber was 3D printed at 0.25 mm resolution using stereolithography and a clear photopolymer resin (Clear Resin V4; Form 2, Formlabs, Somerville, Massachusetts, USA).

Before MRI acquisition, the marmosets were first acclimatized to the head-fixation system and a mock MRI environment (including sequence sounds; see ref. [18] for open-source hardware designs for the animal holder and detailed training procedures). Each marmoset was acclimatized over the course of three weeks prior to imaging.

**Face processing task.** A block design was used in which nine baseline blocks (18 s each) were alternated with eight task blocks (12 s each; see Fig. 5). During baseline blocks, a 0.36° circular black cue was displayed in the center of the screen against a gray background. During task blocks, the dot disappeared and a video was presented in the center of the screen (6° height × 10.6° width)—three stimulus sets were used (counterbalanced between animals), with four pseudo-randomized task conditions each (directed gaze, averted gaze, and scrambled versions of each; see Fig. 5 for representative stimuli). For the directed and averted-gaze videos, five

marmosets were filmed while they sat non head-fixed in a marmoset chair[27]; 12 s clips were created using custom video-editing software (iMovie, Apple Incorporated, California, USA). Scrambled versions of the videos were created by random rotation of the phase information using a custom program (Matlab, The Mathworks, Matick, MA)—the same random rotation matrix was used for each frame to preserve motion components.

Two of the four monkeys were rewarded at the start and end of every block to keep these animals awake and engaged. The liquid reward (diluted sweetened condensed milk) was delivered via infusion pump (Model NE-510, New Era Pump Systems, Inc., Farmingdale, New York, USA). The reward volume was set to 50 microliters per dispense and was delivered over the course of 1 s; note that reward tube was placed outside of the marmosets mouth (~5 mm away) and thus they needed to extend their tongue in order to lick the reward from the tube. We have previously isolated the reward-related circuitry using a similar block design paradigm[21] and this circuitry is, for the most part, discrete from the face circuitry of interest, as described below.

The stimuli were presented via projector (Model VLP-FE40, Sony Corporation, Tokyo, Japan), reflected from a first surface mirror, which back-projected the image onto a plastic screen that was affixed to the front of the scanner bore. The stimuli were presented via Keynote (Version 7.1.3, Apple Incorporated, California, USA), with the stimulus timing (based on a per image volume repetition time (TR) transistor-transistor logic (TTL) pulse) achieved using a Raspberry Pi (Model 3 B+, Raspberry Pi Foundation, Cambridge, UK) programmed in-house.

**Image acquisition.** An integrated animal holder and 5-channel radiofrequency receive array was used to rigidly fix the animal's head chamber to the receive coil[18]. An MRI-compatible camera (Model 12M-i, MRC Systems GmbH, Heidelberg, Germany) allowed for continuous monitoring by a veterinary technician for any sign of struggle or discomfort. Given that skull-attached chambers are generally accompanied by magnetic-susceptibility image artifacts (via differences in the magnetic susceptibility between the chamber, adhesive, air, and tissue, as well as the surgical displacement of the skin, fat, and muscle), we sought to ameliorate this distortion by filling the chamber with a water-based lubricant gel (MUKO SM321N, Canadian Custom Packaging Company, Toronto, Ontario, Canada) prior to each imaging session[18].

Data were acquired using a 9.4 T 31 cm horizontal bore magnet (Varian/ Agilent, Yarnton, UK) and Bruker BioSpec Avance III console with the software

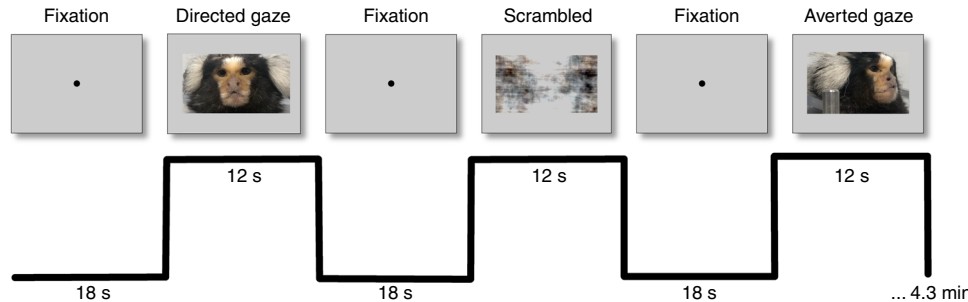

**Fig. 5 Stimuli and experimental design for task-based fMRI.** Top shows the stimuli for the fixation condition and the task conditions, including directed gaze, averted gaze, and scrambled videos. The black line below the stimuli shows the task timing.

package Paravision-6 (Bruker BioSpin Corp, Billerica, MA), a custom-built high-performance 15-cm-diameter gradient coil with 400 mT/m maximum gradient strength (xMR, London, CAN; Peterson et al.[28]), and the receive coil described above. Radiofrequency transmission was accomplished with a quadrature birdcage coil (12-cm inner diameter) built in-house.

**Imaging parameters.** Functional imaging was performed over multiple sessions (days) for each animal, with 6–8 task-based functional runs (at 172 volumes each; each session lasted 30–60 min, including sequence preparations) per animal with the following parameters: TR = 1500 ms, TE = 15 ms, flip angle = 40 degrees, field of view = 64 × 64 mm, matrix size = 128 × 128, voxel size = 0.5 × 0.5 × 0.5 mm, slices = 42, bandwidth = 500 kHz, GRAPPA acceleration factor: 2 (anterior-posterior). T2-weighted structural scans were acquired for each animal during one of the awake sessions with the following parameters: TR = 5500 ms, TE = 53 ms, field of view = 51.2 × 51.2 mm, matrix size = 384 × 384, voxel size = 0.133 × 0.133 × 0.5 mm, slices = 42, bandwidth = 50 kHz, GRAPPA acceleration factor: 2.

**Image preprocessing.** The fMRI data were preprocessed using AFNI[29] and FMRIB/FSL[30]. Raw functional images were converted to NifTI format using dcm2niix[31] and reoriented from the sphinx position using FSL. The images were then despiked (AFNI's 3dDespike) and volume registered to the middle volume of each time series (AFNI's 3dvolreg). The motion parameters from volume registration were stored for later use with nuisance regression. Images were smoothed by a 1.5 mm full-width at half-maximum Gaussian kernel to reduce noise (AFNI's 3dmerge). An average functional image was then calculated for each session and registered (FSL's FLIRT) to each animal's T2-weighted image—the 4D time series data were carried over using this transformation matrix. Anatomical images were manually skull-stripped and this mask was applied to the functional images in anatomical space. The T2-weighted images were then nonlinearly registered to the NIH marmoset brain atlas[32] using Advanced Normalization Tools (ANTs[33]) and the resultant transformation matrices stored for later transformation (see below). The olfactory bulb was manually removed from the T2-weighted images of each animal prior to registration, as it was not included in the template image.

**Task-based fMRI comparisons.** The task timing was convolved to the hemodynamic response (using AFNI's 'BLOCK' convolution) and a regressor was generated for each condition to be used in a regression analysis (AFNI's 3dDeconvolve) for each run. All four conditions were entered into the same model, along with polynomial (N = 5) detrending regressors, bandpass regressors, and the motion parameters derived from the volume registration described above. The resultant regression coefficient maps were then registered to template space using the transformation matrices described above and then converted to Z value maps. The Z value maps for each monkey were then compared at the group level via t-test (AFNI's 3dttest++). To protect against false positives, a clustering method derived from Monte Carlo simulations was applied to the resultant t-test maps (using AFNI's AlphaSim).

**Resting-state seed analysis.** To corroborate the locations of the face patches identified with the task-based analysis, we indexed task-independent functional connectivity of each of the identified temporal face patches described below (i.e., those found by contrasting the directed and averted-gaze conditions from the task-based analysis). To do so, we acquired RS-fMRI from five fully awake adult marmosets (including the four marmosets used above) using the same fMRI acquisition parameters as described above, but with 600 volumes. In total, 35 15 min sessions were acquired and used in this analysis. With this data, we calculated seed-based connectivity across the brain using a 1.5 mm cubic region of interest placed at the center of mass of each face patch in temporal lobes. The preprocessing steps described above were also used for the RS-fMRI seed analysis, apart from the task regressors. Instead, the mean time courses extracted from the four regions of interest were used as the regressors, for each run.

**Comparison with tracer-based cellular connectivity.** With the recent release of tracer-based cellular connectivity maps across marmoset cortex in volume space[19], we were able to directly compare retrograde histochemical tracing in marmosets with our task-based face-selective topologies. Explicitly, we focused on the tracer maps from two injections (CJ164-DY and CJ164-FB; marmosetbrain.org) located most proximally to the cingulate cortex patch found by contrasting the directed and averted-gaze task-based fMRI conditions here (i.e., the cingulate patch sensitive to socially relevant processing). CJ164-DY (diamidino yellow) was injected close to the midline part of area 8b within the right hemisphere of a marmoset yielding 40,628 total labelled cells. CJ164-FB (fast blue) was injected into the midline part of area 8b with slight invasion on area 24c in the right hemisphere of a marmoset yielding 33,785 total labelled cells. We also examined tracer injections into area 45/47 L (CJ73-FE and CJ800-CTBgr). CJ73-FE (fluoro emerald) was injected in 47 L within the right hemisphere of a marmoset yielding 2217 total labelled cells. CJ800-CTBgr (CTB green) was injected in area 45, with some invasion into area 47 L close to the midline part of area 8b within the right hemisphere of a marmoset yielding 26,386 total labelled cells. Note that at the time of this experiment (2020), the temporal injections reported by this resource were relatively sparse and not specific to the face patches found with our fMRI design. To bring the fMRI data and tracer-based connectivity into the same space, the anatomical template (described in 19) was nonlinearly registered to the NIH marmoset brain atlas[32] using ANTs[33]—the volumetric injection data were then brought into template space via this transformation matrix. Note that the tracer data are for the left hemisphere only; as such, we mirrored the left hemisphere data onto both hemispheres.

**Eye tracking.** To investigate differences in patterns of eye movements between conditions we performed eye tracking outside of the scanner (i.e., free of MRI-induced noise, but with identical stimuli; see Fig. 5). Eye positions were digitally recorded at 1 kHz via video tracking of the left pupil (EyeLink 1000, SR Research, Ottawa, ON, Canada). Animals were head restrained in a custom chair[27] mounted to a table in a sound attenuating chamber (Crist Instruments Co., Hagerstown, MD, USA). A spout was placed at the monkey's mouth to deliver reward (acacia gum) via an infusion pump (Model NE-510, New Era Pump Systems, Inc., Farmingdale, New York, USA). In each session, eye position was calibrated by rewarding 300 to 600 ms fixations on dots presented at one of five locations on the display monitor using the CORTEX real-time operating system (NIMH, Bethesda, MD, USA). All stimuli were presented on a CRT monitor (ViewSonic Optiquest Q115, 76 Hz non-interlaced, 1600 × 1280 resolution). A TTL pulse triggered from a photodiode was used to determine the start of each block. Animals were intermittently rewarded at random time intervals to maintain their interest.

Analysis was performed using Python code written in-house. Eye velocity (visual deg/s) was obtained by smoothing and numerical differentiation. Saccades were defined as radial eye velocity exceeding 30 deg/s. Fixations were defined as periods where radial eye velocity remained below 10 deg/s for at least 50 ms. Differences in saccade amplitudes and fixations durations between conditions were analysed with repeated measures analysis of variance (ANOVA) using SPSS (v.25, IBM Corp, 2019) statistical software (post-hoc comparisons were Bonferroni-corrected).

**Reporting summary.** Further information on research design is available in the Nature Research Reporting Summary linked to this article.

## Data availability
The datasets generated during and/or analysed during the current study are available in the https://gin.g-node.org/ repository, https://gin.g-node.org/everling_lab_marmosets/marmoset_face_processing. Source data are provided with this paper.

## Code availability
The code used for the current study are available in the https://gin.g-node.org/ repository, https://gin.g-node.org/everling_lab_marmosets/marmoset_face_processing.

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

## Acknowledgements

Support was provided by the Canadian Institutes of Health Research (FRN 148365, FRN 353372), a Brain Canada Platform Support Grant and the Canada First Research Excellence Fund to BrainsCAN. We wish to thank Cheryl Vander Tuin, Whitney Froese, Kathrine Faubert, and Miranda Bellyou for surgical assistance, animal preparation and care and Dr. Alex Li for scanning assistance.

## Author contributions

D.J.S., J.S., K.D.J., R.S.M., W.A.F., and S.E. designed research. D.J.S., J.S., K.D.J., and S.E. performed research and analysed data. D.J.S. wrote the manuscript. D.J.S., J.S., K.D.J., R.S.M., W.A.F., and S.E. edited the manuscript.

## Competing interests

The authors declare no competing interests.
