## [Peer Review File · Nature Communications]

Reviewers' Comments:

Reviewer #1:

Remarks to the Author:

This paper provides the first functional evidence of face-selective regions of cortex ("patches") in the marmoset monkey. This is the first such evidence in a New World monkey, and demonstrates that even simian primates with small brains have the brain "hardware" associated with using facial information for social interaction. This is important in view of the increased use of marmosets in preclinical research, in suggesting their potential usefulness for the study of conditions such as autism.

The paper is generally sound, and I only have one major suggestion. All other comments are relatively minor.

Major point: Correlation between functional and structural connectivity: Why were only the medial face patches explored in this way? Were there no injections similarly close to the lateral face patches (45/ 47L)? If so, it would be interesting to know if these are also correlated with the face network. At the moment it sounds like only half of the comparison was done, and the reader is left wondering why. If the correlation with the face patch network is less compelling for lateral sites, is there a potential reason, given what is currently known about the function of lateral versus medial sites? could it be that only the medial patch is truly "face selective", with the lateral patch corresponding to a combination of face-selective cells and visuomotor/ attention specific activity? In macaques, isn't area 45 included in the frontal eye field by some authors?

Minor comments and suggestions:

L25: "Primates have evolved the ability transmit important social information" – this is not general of all primates – prosimians for example have limited facial expression. Suggestion: "Many primates have the [delete EVOLVED] ability TO transmit..."

Also, the paper does not really deal with facial expressions, other than gaze direction. This could cause a false expectation among readers. Consider a different introductory sentence.

L58-60: "It is unclear whether a similar network exists in New World primates, who separated ~35 million years ago from Old World primates (Schrager and Russo, 2003) and have a less elaborated frontal cortex (Solomon and Rosa, 2014)". – I would caution against generalizing from marmoset to all New World monkeys. These are a diverse group, and not all "simplified" characteristics that exist in the marmoset necessarily exist in other species. A case in point is the capuchin monkey (*Cebus*), which is in many ways more similar to the macaque than to the marmoset in terms of cortical organization (e.g. Padberg et al. *J Neurosci* 2007; Rosa et al. *Cereb Cortex* 2019). Perhaps the second part of this sentence can be deleted. A possibility would be to include the intended information (that the marmoset and several other new world monkeys have a relatively small frontal lobe) at the end of this paragraph, or elsewhere in the introduction, thus clearly attributing it to this species. The paper by Sneve et al. (*Cereb Cortex* 2019) would be a better reference for this fact.

L118-119: "a fixation point... but also as a visual stimulus that served to mitigate visual nystagmus invoked by the high magnetic field". The possible effects of this artifact need to be better explained – what is the typical amplitude of this eye movement, and how much does the fixation point remedy it. If this has been documented in a previous paper by the group, a reference would be sufficient.

L166-167: "Functional imaging was performed over multiple sessions (days) for each animal, with 6 - 8 task-based functional runs": It would be interesting to know how long the marmosets can stay

engaged in this task. Were they in the scanner for minutes, hours?

L207: "To do so, we acquireD RS-fMRI from five..."

L271-272: these patches are not indicated in Figure 3. It would be better to have an extra row in this figure in which the patches are outlined and labeled, to better show their relationship to the areas indicated in C.

L291-296: The information shown in B would be more easily grasped if the face patches were also indicated in the flat maps, even approximately. At the moment the two types of information (parasagittal sections vs. maps) are difficult to compare. Alternatively, the figure could use 3-d views of the patterns of labeled cells (also available from the marmosetbrain web site) to allow a more prompt comparison with the RSC data in A.

L298: Eye tracking: why were the eye tracking experiments done in a different set of marmosets?

L587-589: I am assuming that the line colors each correspond to one marmoset; please confirm and change figure legend accordingly.

Best regards
Marcello Rosa

Reviewer #2:

Remarks to the Author:

This work titled, Face selective patches in marmoset frontal cortex, reveals two face patches in anterior cingulate cortex and lateral prefrontal cortex, using a state-of-the-art ultra-high field (9.4 Tesla) fMRI. The result is further strengthened by the functional connectivity study by resting state fMRI and the structural connectivity via cellular-level tracing (not done in this work, but compared to the result in a recent work, Majka et al. 2020). The finding is quite simple but could have an important impact, as it demonstrated that one of the New World primates, marmoset, possesses a similar face processing network to that of the macaque. This has two-fold importance. First, it provides supporting evidence that the face processing network evolved before the separation of Old World and New World monkeys ~35 million year ago. And, second, this result gives researchers a stronger reason to start working with marmoset, as it is easier to breed and having a shorter life cycle, and potentially more molecular/genetic tools could be applied. Overall, I find the work interesting, though it may be a bit short in content and there may be some confounding factors that need to be explained further. I am in favor to recommend for publication, with the following concerns being answered.

1. Is it the profile view or the averted gaze?

In Freiwald et al's 2010 paper, "Functional Compartmentalization and Viewpoint Generalization Within the Macaque Face-Processing System", it shows that the neurons in different face patches in macaque respond differently to different viewing angles of the face image (e.g. frontal vs. profile view). It was a single unit level study, so not sure how much does that apply to the fMRI level. However, I find it not very convincing from the illustration in the paper referring to the frontal view as directed gaze, and the profile view as averted gaze. It was more convincing in Shepherd and Freiwald, 2018 in the macaque experiment. I wonder if the authors could explain if this is a possible confound or they have reason to believe this is more about social context rather than just viewing angle (more geometric explanation). Minor point: would be nice to see the stimuli video to understand what exactly is presented to the animal.

2. In Tsao et. al's 2008 paper of "Patches of face-selective cortex in the macaque frontal lobe", they identified 3 face patches in the frontal cortex. PA -- prefrontal arcuate, PO in lateral orbital sulcus and PV in the infrapincipal dimple. However, in this work, the authors found the patches in the lateral prefrontal cortex, which may correspond to PA and/or PV. However, how does the anterior cingulate cortex one correspond to in macaque? And, is there any clue that there may be a patch in orbitofrontal cortex (even maybe not statistically significant). I think it would be interesting to discuss how these patches in marmoset could relate to the ones in the macaque face processing network. I do understand it is dangerous to extrapolate too much to consider a 1 to 1 match between the two species though.

3. When doing the resting state fMRI experiment. I wonder what would happen if using the frontal face patches as the seed. It may reveal a bit more useful information from the perspective originating from the frontal face patches, and it correlates to the cellular-level tracing experiment better. Also, the tracing experiment section seems a bit light in the current manuscript. For example, is the tracer antegrade or retrograde (or both?). I don't think it is described.

Reviewer #3:

Remarks to the Author:

Assessing face selectivity by comparing videos of directed-gaze faces to scrambled videos does not seem to me to be face specific. I would have preferred to see the contrast be some kind of object video. Comparing directed vs averted gaze video also was not that impressive because the parts activated look quite similar to me, just slightly lower in the averted condition. Since the eyes were not visible (I assume) in the averted gaze condition, that's enough to explain the lower activation, since eyes are an important part of the face.

We would like to thank the editors and reviewers for their time and effort on our manuscript. We are grateful for the refinements offered by the reviewers – the changes to the text and additional analyses (e.g., tracer and seed-based analyses of area 45/47L) have very much improved the quality of our manuscript. Below, please find each critique listed with a detailed response. Corresponding changes in the manuscript are indicated in red font.

REVIEWER COMMENTS

Reviewer #1 (Remarks to the Author):

This paper provides the first functional evidence of face-selective regions of cortex ("patches") in the marmoset monkey. This is the first such evidence in a New World monkey, and demonstrates that even simian primates with small brains have the brain "hardware" associated with using facial information for social interaction. This is important in view of the increased use of marmosets in preclinical research, in suggesting their potential usefulness for the study of conditions such as autism.

The paper is generally sound, and I only have one major suggestion. All other comments are relatively minor.

Reply: Thank you. We appreciate the positive summary and constructive comments.

Major point: Correlation between functional and structural connectivity: Why were only the medial face patches explored in this way? Were there no injections similarly close to the lateral face patches (45/ 47L)? If so, it would be interesting to know if these are also correlated with the face network. At the moment it sounds like only half of the comparison was done, and the reader is left wondering why. If the correlation with the face patch network is less compelling for lateral sites, is there a potential reason, given what is currently known about the function of lateral versus medial sites? could it be that only the medial patch is truly "face selective", with the lateral patch corresponding to a combination of face-selective cells and visuomotor/ attention specific activity? In macaques, isn't area 45 included in the frontal eye field by some authors?

Reply: This is a very good point. As suggested, we have now included tracer injections into areas 45 and 47L. As shown in the figure below (Figure 4c in the manuscript), these injections, particularly CJ800-CTBgr (area 45) is very strongly connected to the face patches. In addition, this pattern overlaps well with the RS functional connectivity seed from area 45/47 (Figure 4A, bottom; also added to this figure as per Reviewer 2's suggestion). We have added the following to the text regarding these injections:

Lines 305-312: "Further, consonant with our functional connectivity analysis, anterior cingulate cortex injections show strong connectivity with the anterior face patches (AD and MD), but weaker connectivity with the posterior face patches (PD and PV). As shown in Figure 4C, injections into area 45/47 (particularly CJ800-CTBgr) show very strong connectivity with both anterior cingulate cortex and also along the occipitotemporal pathway harboring the face patches. Accordingly, as also suggested by the resting-state seed analysis above, the 8b/24 patch is

strongly connected to the anterior faces patches (AD and MD) whereas the lateral patch (45/47L) is more broadly connected across the face patches.”

(A) RS-fMRI functional connectivity

(B) Tracer-based structural connectivity

Regarding the lateral patch (45/47L), yes, it is very well possible that this patch could be a combination of both face-selective cells and those involved in visuomotor activity, especially at the rostral end of 45. We have recently localized the FEF in marmosets directly with task fMRI (Schaeffer et al., 2019) and microsimulation (Selvanayagam et al., 2019). From these studies, however, we believe that the FEF is dorsal and medial to the patch of the current study. We have summarized our current evidence for this in the figure below, with the location of the 45/47L patch indicated by green arrows.

Microstimulation evoking saccades (Selvanavagam et al., 2019)

Minor comments and suggestions:

L25: “Primates have evolved the ability transmit important social information” – this is not general of all primates – prosimians for example have limited facial expression. Suggestion: “Many primates have the [delete EVOLVED] ability TO transmit...”

Also, the paper does not really deal with facial expressions, other than gaze direction. This could cause a false expectation among readers. Consider a different introductory sentence.

Reply: Thank you. We’ve removed this sentence from the manuscript.

L58-60: “It is unclear whether a similar network exists in New World primates, who separated ~35 million years ago from Old World primates (Schrage and Russo, 2003) and have a less elaborated frontal cortex (Solomon and Rosa, 2014)”. – I would caution against generalizing from marmoset to all New World monkeys. These are a diverse group, and not all “simplified” characteristics that exist in the marmoset necessarily exist in other species. A case in point is the capuchin monkey (Cebus), which is in many ways more similar to the macaque than to the marmoset in terms of cortical organization (e.g. Padberg et al. J Neurosci 2007; Rosa et al. Cereb Cortex 2019). Perhaps the second part of this sentence can be deleted. A possibility would be to include the intended information (that the marmoset and several other new world monkeys have a relatively small frontal lobe) at the end of this paragraph, or elsewhere in the introduction, thus clearly attributing it to this species.

The paper by Sneve et al. (Cereb Cortex 2019) would be a better reference for this fact.

Reply: As suggested, we have removed the latter half of this sentence from both the aforementioned lines (L58-60) and also from the abstract. We have also added the following sentence to toward the end of the introductory paragraph, citing Sneve et al. (2019):

Lines 63-64: “Marmosets, however, have a less elaborated frontal cortex when compared to Old World primate species including macaques (Sneve et al., 2019).”

L118-119: “a fixation point... but also as a visual stimulus that served to mitigate visual nystagmus invoked by the high magnetic field”. The possible effects of this artifact need to be better explained – what is the typical amplitude of this eye movement, and how much does the fixation point remedy it. If this has been documented in a previous paper by the group, a reference would be sufficient.

Reply: We have not overtly measured adaptation to nystagmus in the scanner. This is difficult to do, as the marmosets often close their eyes in the absence of a visual stimulus. Closing their eyes seems to make the nystagmus worse, as evidenced by acute heavy nystagmus upon reopening that decreases as the eyes remain open. Generally, in most marmosets, nystagmus is strongest during the first MRI session (which we typically use to acquire non-functional sequences), then decreases dramatically within 20-30 minutes during the first session with light (< 5 degrees). As such, by the time the marmosets actually perform the task (e.g., after training and adaptation), the nystagmus has adapted. Further, if present (albeit minimal), this effect would be present in both the baseline and task conditions. At the risk of over-complicating this issue, we have removed this sentence from the manuscript as it is more relevant to training than it is a task confound.

L166-167: “Functional imaging was performed over multiple sessions (days) for each animal, with 6 - 8 task-based functional runs”: It would be interesting to know how long the marmosets can stay engaged in this task. Were they in the scanner for minutes, hours?

Reply: We apologize this was not clearer. Typically, they were engaged in the task for ~30 minutes and in the MRI for ~45-60 minutes per session (including time for shimming, B0 mapping, and other preparations). As such, to get a full data set on each animal, we collected the data over two to three sessions (days). We have clarified this in the manuscript as follows:

Lines 166-168: “Functional imaging was performed over multiple sessions (days) for each animal, with 6 – 8 task-based functional runs (at 172 volumes each; each session lasted 30-60 minutes, including sequence preparations) per animal with the following parameters:”

L207: “To do so, we acquireD RS-fMRI from five...”

Reply: Thank you. Corrected.

L271-272: these patches are not indicated in Figure 3. It would be better to have an extra row in this figure in which the patches are outlined and labeled, to better show their relationship to the areas indicated in C.

Reply: As suggested, we have added an extra row to Figure 3 which shows the patch outlines and labels. Please find this figure pasted below.

L291-296: The information shown in B would be more easily grasped if the face patches were also indicated in the flat maps, even approximately. At the moment the two types of information (parasagittal sections vs. maps) are difficult to compare. Alternatively, the figure could use 3-d views of the patterns of labeled cells (also available from the marmosetbrain web site) to allow a more prompt comparison with the RSC data in A.

Reply: Labels have been added to the flat maps. While we have put the patterns of labelled cells on surface maps, we find that using this mid-surface modelling transformation can underrepresent (i.e., hide) the labelled cells in some cases, especially for smaller clusters. This can in part be ameliorated by changing the mapping parameters, but we would prefer not to do this as it can also “induce” some false positives through spatial smoothing – this is apparent on the figure pasted below. Indeed, we agree this is easier to compare (allowing the reader to quickly see that RS-fMRI and tracer injections overlap quite well), but fear that we would misrepresent the tracer data in some cases on the surfaces. For this reason, we decided to show this on non-surface parasagittal slices, which have minimal distortion. We are, however, happy to display the figure as below at the discretion of the reviewer/editors.

L298: Eye tracking: why were the eye tracking experiments done in a different set of marmosets?

Reply: This was simply logistical – one marmoset lost an implant after the MRI experiments, another was euthanized, and another was not trained in the upright chair. The fourth was included in the eye tracking group. The other marmosets who participated in eye tracking were not able to be scanned because their implants were not MRI compatible (radio-opaque dental cement was used).

L587-589: I am assuming that the line colors each correspond to one marmoset; please confirm and change figure legend accordingly.

Reply: Yes, each line color is an individual marmoset. We have changed the Figure caption as follows:

Lines 636-638: “Figure 5. Distributions of saccades by saccade amplitude in visual degrees (A) and fixations by fixation duration in ms (B) for each condition separately for 5 marmoset subjects. Different line colors denote individual marmoset subjects. Vertical red lines represent the group median value.”

*Best regards
Marcello Rosa*

Reviewer #2 (Remarks to the Author):

This work titled, Face selective patches in marmoset frontal cortex, reveals two face patches in anterior cingulate cortex and lateral prefrontal cortex, using a state-of-the-art ultra-high field (9.4 Tesla) fMRI. The result is further strengthened by the functional connectivity study by resting state fMRI and the structural connectivity via cellular-level tracing (not done in this work, but compared to the result in a recent work, Majka et al. 2020). The finding is quite simple but could have an important impact, as it demonstrated that one of the New World primates, marmoset, possesses a similar face processing network to that of the macaque. This has two-fold importance. First, it provides supporting evidence that the face processing network evolved

before the separation of Old World and New World monkeys ~35 million year ago. And, second, this result gives researchers a stronger reason to start working with marmoset, as it is easier to breed and having a shorter life cycle, and potentially more molecular/genetic tools could be applied. Overall, I find the work interesting, though it may be a bit short in content and there may be some confounding factors that need to be explained further. I am in favor to recommend for publication, with the following concerns being answered.

Reply: Thank you – we appreciate your support of this work and helpful comments. Please find detailed responses below.

1. Is it the profile view or the averted gaze?

In Freiwald et al's 2010 paper, "Functional Compartmentalization and Viewpoint Generalization Within the Macaque Face-Processing System", it shows that the neurons in different face patches in macaque respond differently to different viewing angles of the face image (e.g. frontal vs. profile view). It was a single unit level study, so not sure how much does that apply to the fMRI level. However, I find it not very convincing from the illustration in the paper referring to the frontal view as directed gaze, and the profile view as averted gaze. It was more convincing in Shepherd and Freiwald, 2018 in the macaque experiment. I wonder if the authors could explain if this is a possible confound or they have reason to believe this is more about social context rather than just viewing angle (more geometric explanation). Minor point: would be nice to see the stimuli video to understand what exactly is presented to the animal.

Reply: Although we did not expect to disentangle the effects found in Freiwald and Tsao (2010) (that of cellular-level tuning) using fMRI, the present results do not necessarily contradict those found in the Freiwald 2010 study. The Freiwald 2010 study demonstrated the tuning of individual cells to head orientation rather than a more global effect of population basis, which would potentially be detectable with fMRI. Although it may be possible to detect this effect with fMRI using specific adaptation paradigms, we do not believe that a this is the case for the frontal patches – indeed, Shepherd and Freiwald (2018) used a conspecific version of the same task in macaques (i.e., the same species as the 2010 electrophysiology study) and did not see a modulation effect of head orientation when using fMRI.

Regarding directed and averted gaze views, yes, like in Shepherd and Freiwald (2018) the eye(s) (at least one) were visible in the averted gaze condition with the camera set up for ~45 degree head rotation (albeit sometimes more or less as the marmosets were not head-fixed and free to make movements). We have replaced the "averted gaze" condition image with a more representative video frame to avoid confusion (Figures 1 and 3). Even when more toward the "profile" (i.e., 90 degree) view, we believe that this is still an "averted gaze" condition, such that it is a differential social context from the directed gaze condition that is recognizable from feature differences. In addition to providing the stimuli videos (link below), we have also uploaded gaze patterns for the different conditions. As shown in the "averted.mp4" video, the marmosets could indeed see the facial features (e.g., eye(s)) of the marmoset in the video (even when >45 degrees) which they also fixated on, as shown by quantifying fixations (see Figure pasted below).

Thus, although we do agree that the temporal face patches can indeed be orientation selective when measured at a cellular level, we hypothesize that this is not the case for the frontal patches (Freiwald and Tsao (2010) did not record from the frontal patches in macaques) given that the anterior cingulate patch, for example, would have to devote a majority of neurons to coding frontal views as visual features. We know that temporal face patches are already tuned to frontal views and thus serial coding of head orientation would be unlikely. That being said, we do fully agree that this is an important question that should be probed. Perhaps this study, which is the first to localize the frontal patches in this species, could serve as a starting block from which electrophysiological experiments could hone in on this potential effect.

https://gin.g-node.org/everling_lab_marmosets/marmoset_face_processing/

2. In Tsao et. al's 2008 paper of "Patches of face-selective cortex in the macaque frontal lobe", they identified 3 face patches in the frontal cortex. PA -- prefrontal arcuate, PO in lateral orbital sulcus and PV in the infrapincipal dimple. However, in this work, the authors found the patches in the lateral prefrontal cortex, which may correspond to PA and/or PV. However, how does the anterior cingulate cortex one correspond to in macaque? And, is there any clue that there may be a patch in orbitofrontal cortex (even maybe not statistically significant). I think it would be interesting to discuss how these patches in marmoset could relate to the ones in the macaque face processing network. I do understand it is dangerous to extrapolate too much to consider a 1 to 1 match between the two species though.

Reply: In terms of cytoarchitectonic location, the anterior cingulate patch is very similar between marmosets and macaques. In both species, this patch seems to reside just dorsal to area 32, extending into areas 24, 8, and 9. Please find the figure pasted below which summarizes relevant task fMRI and RS-fMRI results across the literature between the two species as they relate to cytoarchitecture. We have added the following text to directly discuss this frontal patch as it relates to what has been found in macaques:

Lines 360-365: "As previously demonstrated in macaques (using a similar conspecific video set), the patch in anterior cingulate cortex (8b/24) was specific to the directed gaze condition, suggesting that this patch is involved in socially relevant face processing (Shepherd and Freiwald, 2018). The location of this patch is also in line with what is found in macaques,

residing just dorsal to area 32, extending into areas 24, 8, and 9 (Schwiedrzik et al., 2015; Shepherd and Freiwald, 2018).”

Macaque

Marmoset

Regarding the orbitofrontal region, yes, we did indeed see differential activation for the directed gaze condition in area 13 of OFC. As stated previously in our radiofrequency coil hardware manuscript (Schaeffer et al., 2019, *NeuroImage*), however, our custom receive coil is less sensitive to signal in orbitofrontal cortex (also related to proximity of the large, fluid filled eyes). As shown in the figure below, these orbitofrontal regions reside within an area of relatively low signal-to-noise ratio (SNR). As such, although a promising result, we are less confident in these findings than the other frontal face patches. Nonetheless, we have added this to the results, along with the caveat about lower signal in this area.

Lines 268-270: “In frontal cortex, the social videos showed peaks laterally in 45/47L and orbitofrontally in 13L (albeit orbitofrontal cortex suffered from relatively low signal-to-noise ratio, see Schaeffer et al., 2019a).”

3. When doing the resting state fMRI experiment. I wonder what would happen if using the frontal face patches as the seed. It may reveal a bit more useful information from the perspective originating from the frontal face patches, and it correlates to the cellular-level tracing experiment better. Also, the tracing experiment section seems a bit light in the current manuscript. For example, is the tracer antegrade or retrograde (or both?). I don't think it is described.

Reply: Yes, thank you, this is a good idea. As pasted below, we have added RS-fMRI seed analyses for the frontal patches to Figure 4. These patterns indeed showed very good correspondence to the tracer injections into these areas (please also see reply to Reviewer 1's comments above regarding the addition of LFC tracers to Figure 4 and specific text additions).

As we found with connectivity of the temporal lobe patches, the frontal anterior cingulate patch seems to be connected more strongly to the anterior temporal patches, whereas the lateral frontal patch was more broadly connected across the face patches along the occipitotemporal axis.

These tracers are all retrograde only. As requested, we have added the following details to the methods section regarding the tracer injections:

Lines 224-231: “With the recent release of tracer-based cellular connectivity maps across marmoset cortex in volume space (Majka et al., 2020), we were able to directly compare retrograde histochemical tracing in marmosets with our task-based face selective topologies. Explicitly, we focused on the tracer maps from two injections (CJ164-DY and CJ164-FB; marmosetbrain.org) located most proximally to the cingulate cortex patch found by contrasting the directed and averted gaze task-based fMRI conditions here (i.e., the cingulate patch sensitive to socially-relevant processing). CJ164-DY (diamidino yellow) was injected close to the midline part of area 8b within the right hemisphere of a marmoset yielding 40,628 total labelled cells. CJ164-FB (fast blue) was injected into the midline part of area 8b with slight invasion on area 24c in the right hemisphere of a marmoset yielding 33,785 total labelled cells. We also examined tracer injections into area 45/47L (CJ73-FE and CJ800-CTBgr). CJ73-FE (fluoro emerald) was injected into 47L within the right hemisphere of a marmoset yielding 2,217 total labelled cells. CJ800-CTBgr (CTB green) was injected in area 45, with some invasion into area 47L close to the midline part of area 8b within the right hemisphere of a marmoset yielding 26,386 total labelled cells.”

(A) RS-fMRI functional connectivity

(B) Tracer-based structural connectivity

Reviewer #3 (Remarks to the Author):

Assessing face selectivity by comparing videos of directed-gaze faces to scrambled videos does not seem to me to be face specific. I would have preferred to see the contrast be some kind of object video. Comparing directed vs averted gaze video also was not that impressive because the parts activated look quite similar to me, just slightly lower in the averted condition. Since the eyes were not visible (I assume) in the averted gaze condition, that's enough to explain the lower activation, since eyes are an important part of the face.

Reply: Thank you for reviewing our manuscript. We agree that contrasting faces with objects is a valuable comparison – indeed, a comparison that has been conducted in marmosets already. Hung et al. (2015, *Journal of Neuroscience*) used task-based fMRI to compare photos of marmoset faces with objects and found that the face sensitive patches (AD and MD) were more

responsive to faces than objects – a pattern similar to that found with phase scrambled images of faces. The main purpose of our study was slightly more specific – to contrast differential social contexts of faces to determine if these socially relevant feature differences result in differential activation. Please note that our task design was a direct adaptation of an already published effect in macaques (Shepherd and Freiwald, 2018, *Neuron*) in which the directed versus averted gaze videos result in differential activation in frontal cortex. We simply modified this task for conspecific use in marmosets.

As described in response to Reviewer 2 above, the averted condition not only contained the eye(s), but the marmosets fixated on this feature as shown in the fixation map above. Further, although the averted gaze condition does generally show less activation (e.g., in the temporal patches) than the directed gaze condition, the averted gaze condition topology is not just a scaled down version of the directed gaze condition, particularly in the frontal patches. This becomes more apparent when the maps are shown unthresholded. As shown in the figure pasted below, the averted gaze condition did yield activation in the temporal face patches (i.e., evidence that the marmosets were processing the faces), but there was *no activation* in the anterior cingulate face patch. As such, we feel that the data strongly supports the conclusion that social condition (directed gaze) leads to a differential pattern of activation from the averted gaze condition and that these differences are related to socially relevant feature differences.

Reviewers' Comments:

Reviewer #1:

Remarks to the Author:

The authors have addressed all substantial concerns raised in my review. I am happy to recommend the revised paper for publication.

I have only one minor correction to suggest: in the revised abstract, line 41: Catarrhini not Catarrhine (to match usage of Platyrrhini).

Reviewer #2:

Remarks to the Author:

I would like to thank the authors to write a clear point-to-point replies to my concerns. I've read the authors' replies to all three reviewers, and was quite satisfied.

For my concerns,

1. Is it the profile view or the averted gaze?

I think the new example image of the averted gaze is more compelling now. And, the authors do admitted that it is still possibly a confounding factor. Though, it is less likely the frontal lobe face patches are coding for head orientation.

2. Compare the frontal lobe face patches location to those in macaque. I think the correspondence is pretty nice. Also, the authors added the possible observation of the patch at the orbitofrontal cortex, with the caveat of low signal-to-noise ratio there.

3. When doing the resting state fMRI experiment. I wonder what would happen if using the frontal face patches as the seed.

In this revised manuscript, the authors have done a more comprehensive resting state analysis using different seeds. Also, some more tracer injections results are added based on suggestions from reviewer #1.

Overall, I think the paper after revision is more complete now, and I would recommend for publication.

****REVIEWERS' COMMENTS:**

Reviewer #1 (Remarks to the Author):

The authors have addressed all substantial concerns raised in my review. I am happy to recommend the revised paper for publication.

Reply: Thank you.

I have only one minor correction to suggest: in the revised abstract, line 41: Catarrhini not Catarrhine (to match usage of Platyrrhini).

Reply: We have replaced “Catarrhine” with “Catarrhini” in the abstract.

Reviewer #2 (Remarks to the Author):

I would like to thank the authors to write a clear point-to-point replies to my concerns. I've read the authors' replies to all three reviewers, and was quite satisfied.

Reply: Thank you, we appreciate your careful review and are glad that we were able to fully address all of your concerns, as you nicely summarize below.

For my concerns,

1. Is it the profile view or the averted gaze?

I think the new example image of the averted gaze is more compelling now. And, the authors do admitted that it is still possibly a confounding factor. Though, it is less likely the frontal lobe face patches are coding for head orientation.

2. Compare the frontal lobe face patches location to those in macaque. I think the correspondence is pretty nice. Also, the authors added the possible observation of the patch at the orbitofrontal cortex, with the caveat of low signal-to-noise ratio there.

3. When doing the resting state fMRI experiment. I wonder what would happen if using the frontal face patches as the seed.

In this revised manuscript, the authors have done a more comprehensive resting state analysis using different seeds. Also, some more tracer injections results are added based on suggestions from reviewer #1.

Overall, I think the paper after revision is more complete now, and I would recommend for publication.